# Comparison of Oral Microbial Composition and Determinants Encoding Antimicrobial Resistance in Dogs and Their Owners

**DOI:** 10.3390/antibiotics12101554

**Published:** 2023-10-20

**Authors:** Laura Šakarnytė, Rita Šiugždinienė, Judita Žymantienė, Modestas Ruzauskas

**Affiliations:** 1Microbiology and Virology Institute, Lithuanian University of Health Sciences, 44307 Kaunas, Lithuania; laura.sakarnyte@lsmu.lt (L.Š.); rita.siugzdiniene@lsmu.lt (R.Š.); 2Department of Anatomy and Physiology, Veterinary Academy, Lithuanian University of Health Sciences, 44307 Kaunas, Lithuania; judita.zymantiene@lsmu.lt

**Keywords:** bacteria, zoonoses, antimicrobial resistance, pets, microbiome, resistome

## Abstract

Consolidated studies on animal, human, and environmental health have become very important for understanding emerging zoonotic diseases and the spread of antimicrobial resistance (AMR). The aim of this study was to analyse the oral microbiomes of healthy dogs and their owners, including determinants of AMR. Shotgun metagenomic sequencing detected 299 bacterial species in pets and their owners, from which 70 species were carried by dogs and 229 species by humans. Results demonstrated a unique microbial composition of dogs and their owners. At an order level, Bacteroidales were the most prevalent oral microbiota of dogs with significantly lower prevalence in their owners where Actinomycetales and Lactobacillales predominated. *Porphyromonas* and *Corynebacterium* were the most prevalent genera in dogs, whereas *Streptococcus* and *Actinomyces* were in animal owners. The resistances to macrolides, tetracyclines, lincosamides and Cfx family A class broad-spectrum β-lactamase were detected in both animal and human microbiomes. Resistance determinants to amphenicols, aminoglycosides, sulphonamides, and quaternary ammonium compounds were detected exceptionally in dogs. In conclusion, the study demonstrated different bacterial composition in oral microbiomes of healthy dogs without clinical signs of periodontal disease and their owners. Due to the low numbers of the samples tested, further investigations with an increased number of samples should be performed.

## 1. Introduction

Zoonotic infections are caused by a wide variety of microorganisms and are transmitted naturally from animals to humans and vice versa. Recently, it was indicated that more than 60% of human pathogens are zoonotic in origin [1]. The One Health concept is focused on consequences, responses, and actions at the animal–human–ecosystem interfaces, especially for emerging zoonoses [2,3] and antimicrobial resistance (AMR), which can spread between humans and animals [4]. A collaborative and multi-disciplinary approach, cutting across boundaries of animal, human, and environmental health, is needed to understand the ecology of each emerging zoonotic disease and the spread of AMR determinants to undertake a risk assessment, and to develop plans for response and control [5]. 

Periodontitis is an infection caused by microorganisms that damage the gingiva and, without treatment, can cause bone loss around teeth, which may lead to tooth loss. This pathology is widespread in humans and especially in dogs, which is reported to be the most common disease affecting dogs (and cats) worldwide [6,7]. Previously, it was reported that 63% of pet owners have never had their pets’ teeth cleaned [8], suggesting a high risk of bacterial infections in dogs and humans they come into contact with. A study performed in Japan in 2012 concluded that close contact could contribute to the transmission of bacteria such as *Eikenella corrodens, Porphyromonas gulae*, *Treponema forsythia*, and *T. denticola* prevalent in dog’s mouths to the owners who had close contact with their animals [9]. A recent study in China also stressed close contact as the major factor for possible transmission of *P. gulae* from dogs to their owners [10]. Some of the above-mentioned bacteria can be found in the healthy microbiome of dogs [11,12]. However, the transmission possibility followed by the prolonged carriage in humans infected through healthy animals is still unclear and needs investigation. Dogs are a major reservoir for zoonotic infections [13]. Zoonotic diseases can be transmitted to humans by infected saliva, aerosols, faeces and urine. Importantly, it is impossible to escape direct and close contact with dogs living in households, as they become family members, live in the same space and often sleep with their owners. Equally, dogs have the same risk of obtaining microbiota from humans. Dog’s mouths have hundreds of bacterial species and pose a risk for their owners [14]. At the beginning of this decade, up to 80% of the bacterial taxa from dogs were unnamed [14]; this knowledge has now been expanded. 

In the last few years, molecular diagnostic methods have experienced rapid development and played an increasingly important role in the diagnostics of bacterial diseases and microbiome studies [15]. These methods have reduced the time from receiving the sample to the final result, and made it possible to detect non-culturable pathogens. 16S rRNA gene sequencing was applied in many studies in which universal PCR primers have been used; the resulting partial 16S gene amplicons, encompassing hypervariable regions, were used to infer taxonomic identifications based upon bioinformatics alignments against sequence databases [16]. The 16S rRNA gene is used extensively in bacterial phylogenetics, species delineation, and now widely in microbiome studies. However, the gene suffers from intragenomic heterogeneity, and reports of recombination and an unreliable phylogenetic signal are accumulating [17]. Full-length sequencing has resulted in much better possibilities to identify bacteria up to the species level [18]. Current microbiological analysis in medicine and veterinary medicine requires fast and appropriate answers about pathogens, antimicrobial susceptibility and virulence factors. Shotgun metagenomic sequencing allows the evaluation of the ecological-level dynamics of AMR and virulence, in conjunction with microbiome analysis [19,20]. 

As bacteria from close environments can share plasmids and other mobile genetic elements, it is important to investigate genes encoding AMR and virulence in zoonotic and other natural microbiota, which can spread between humans and animals. 

Recent metagenomic studies reported that in the genome of potentially pathogenic *Bacteroides*, *Capnocytophaga*, *Corynebacterium*, *Fusobacterium*, *Pasteurella*, *Porphyromnas*, *Staphylococcus* and *Streptococcus* species prevalent in dogs, AMR genes’ encoding resistance to different antimicrobial classes, including those that are critically and highly important for humans, can be identified [21]. 

Beneficial microorganisms are very important for the immunity status of the host [22,23]. These organisms ensure a lower prevalence of pathogenic microbiota but can also be carriers of AMR genes. From this point of view, the microbiomes of healthy animals should also be investigated. Moreover, most of the probiotics used are known as culturable bacteria, but unculturable microorganisms are predominant and, therefore, play an essential role in different physiological functions of animals. There is a lack of information on whether probiotic bacteria can spread from dogs to their owners and vice versa.

Previous studies of dogs’ oral microbiome were mostly based on the detection of microbial pathogens in diseased dogs, particularly with periodontitis [24,25,26,27], although some data about the microbiota of healthy dogs have also been published [28,29]. The data, however, differ according to the separate studies. For example, analysis from a few studies regarding the composition of oral microbiomes in dogs indicated that the most prevalent bacterial genera were *Actinomyces*, *Campylobacter*, *Corynebacterium*, *Haemophilus*, *Lampropedia*, *Neisseria*, *Pasteurella*, *Porphyromonas*, *Rothia* and *Streptococcus* [7,30]. In contrast, in other studies, *Capnocytophaga*, *Flavobacterium*, *Gemella*, *Abiotrophia*, *Frederiksenia* and others [31], or *Paludibacter*, *Prevotella*, *Desulfomicrobium*, *Moraxella*, *Euzebya*, *Proteocatella*, *Fusibacter*, *Leptotrichia* and others [32] were found. More data are needed to obtain more knowledge about the microbial composition in dogs. Comparative data about the microbiome of dogs and their owners sharing the same environment could help understand microbial species interchange between humans and pets and the possible risks of zoonotic agent transfer and the spread of AMR. The aim of this study was to analyse oral microbiomes of healthy dogs and their owners, including determinants of AMR, to obtain more data on possible microbial and AMR transfer between dogs and their owners.

## 2. Results

### 2.1. Microbial Composition of Oral Microbiomes of Pets and Their Owners

Neither viruses nor eukaryotic organisms except the host DNA were detected in the samples obtained from the six dogs and their owners. Bacterial compositions at a phylum and an order level of oral microbiomes of dogs and their owners are presented in Figure 1.

Data from Figure 1 show different prevalence rates of bacteria at a phylum level in dogs and their owners. The most prevalent bacteria in dogs were Bacteroidota, whereas in their owners, Actinobacteria and Firmicutes were the most prominent. Different bacterial compositions were also observed in dogs and their owners at an order level. In this case, Bacteroidales were the most prevalent in dogs. In contrast, Actinomycetales and Lactobacillales were the most prevalent in their owners. Figure 2 displays the data on the microbial composition of oral microbiomes of dogs and their owners at a genus level. 

As can be seen from Figure 2, the most prevalent genera in dogs were *Porphyromonas* (35.5%), followed by *Corynebacterium* (14.9%), *Lampropedia* (9.6%), *Tennerella* (4.4%) and *Arachnia* (4.0%). In humans the most prevalent genera were *Streptococcus* (33.4%), *Actinomyces* (28.1%), *Veillonella* (4.5%) and *Rothia* (4.1%). Although dogs and their owners carried some bacteria of the same genera, the prevalence (number) of the same genus differed significantly among all genera (*p* < 0.05).

In total, 299 bacterial species were detected in the oral microbiomes of pets and their owners; 70 species were carried by dogs, and 229 by humans, demonstrating a much higher alpha diversity in the human oral cavity. The same species of bacteria were not found in the oral cavity of dogs and their owners.

Bacterial species of the most prevalent genera detected in dogs are presented in Figure 3 and Figure 4, whereas humans are in Figure 5, Figure 6 and Figure 7, and are described below. 

The data showed that different species of *Porphyromonas* are prevalent in dogs compared to their owners (Figure 3). In dogs, *P. gulae*, *P. gingivalis*, *P. canoris* and *P. gingivicanis* were the most prevalent. In humans only two species of *Porphyromonas* were detected (*P. pasteri* and *P. catoniae*), which were absent in dogs. 

Figure 4 demonstrates *Corynebacterium* spp. variety, which was the second most prevalent genus in dogs. The data show different *Corynebacterium* species prevalence in dogs, where three species (*C. canis*, *C. freiburgense* and *C. mustelae*) were detected. In contrast, other species (*C. matruchotii* and *C. durum*) were found in humans. 

A plethora of *Streptococcus* species were abundant in dog owners, with the most common prevalence of normal human oral microbiota including *S. oralis*, *S. mitis*, *S. sanguinis*, and *S. infantis*. At the same time no streptococci were detected in the mouth of their dogs. 

*Actinomyces* were the second most abundant taxon in pet owners, with the most common species being *A. oris*, *A. massiliensis*, *A. neuslundii* and others. 

In contrast, in dogs’ oral cavities, only two species were present, including *A. bowdenii* and *A. weisii*, which were not detected in humans (Figure 5). A similar situation was in the case of the well-known bacteria genus *Neisseria*, which was prevalent in both humans and dogs. Still, the species differ substantially, with the most prevalent *N.dumasiana*, *N. canis, N. animaloris* and *N. zoodegmatis* in dogs and *N. mucosae*, *N. elongata, N. sicca* and *N. cinerea* in their owners (Figure 6).

Although *Pauljensenia* species were detected in both samples in quite high amounts (5.0% and 2.1% in owners and dogs, respectively), the species content also differed among the dogs and their owners (Figure 7). 

When analysing all the rest of the species from all the bacterial compositions of the dog microbiomes, we did not find any of the same species to be prevalent in the owners of their pets. All the species prevalent in the microbiomes of dogs and their owners are presented in Appendix A.

### 2.2. Prevalence of AMR Genes

AMR genes detected in bacteria of the oral cavities of dogs and their owners are presented in Table 1. 

Thirty-seven different genetic determinants were detected in the microbiota of dogs and their owners, encoding resistance to different classes of antimicrobials. The most widespread encoded resistances were detected towards β-lactams, tetracyclines and macrolides. Overall the number of similar determinants detected in dogs and their owners was quite low: there were only single genes encoding resistance to β-lactams (*cfx*A), macrolides (*Erm*X) and streptogramins, pleuromutilins and lincomycin class (*Lsa*(C)); however, much more similar determinants were detected to tetracyclines including different ribosomal protection proteins as well as the ABC transporter *Tet(46)* responsible for the efflux. The highest amount of AMR genes in dogs was detected for encoding resistance to tetracyclines (*n* = 280). In contrast, in humans, the most prevalent were genes encoding resistance to macrolides (*n* = 2412), tetracyclines (*n* = 935) and β-lactams (*n* = 196). AMR genes encoding resistance to amphenicols, aminoglycosides and sulphonamides were detected exceptionally in dogs’ microbiota. Dogs also carried bacteria resistant to quaternary ammonium compounds. 

## 3. Discussion

### 3.1. Human Microbiome

In recent years, much attention has been paid to human microbiome studies. One of the most known studies was the Human Microbiome Project (HMP), which aimed to characterise the ‘healthy’ human microbiome as a baseline for reference and comparison studies [33]. Recent studies demonstrated the course of colonisation by microorganisms of the human oral tract, which usually starts as intrauterine environment colonisation, specifically in the amniotic fluid of pregnant women [34]. The baby comes into contact with the microbiota of the uterus and vagina of the mother during delivery, and later with the microorganisms of the atmosphere at birth [35]. The oral cavity later is regularly inoculated with microorganisms from the first feeding onward, and resident oral microbiota acquisition begins [36]. While understanding the factors involved in shaping the human gut microbiome is rapidly progressing, there is still a lack of knowledge regarding dogs [37]. Similarly, in humans it was initially admitted that the gastrointestinal tract of mammals is sterile during the intrauterine foetal life, with the inoculation of microorganisms occurring through contact with the mother’s vagina, skin and ingestion of milk within the first hours following parturition [38]. This assessment was recently challenged due to the detection of bacteria in the placenta, uterus or amniotic fluid in different mammals, with transmission of bacteria from the mother to the foetus potentially in utero [39].

Initial colonisers immediately after birth in humans include *Streptococcus salivarius*, other *Streptococcus* spp., *Lactobacillus, Actinomyces, Neisseria* and *Veillonella*. Later, the microbial community in healthy mouths is conserved, and diversity in the microbiome at a genus level is individual-specific despite some similarities [35]. The genera that usually are prevalent in the healthy human oral microbiome include *Streptococcus*, *Peptostreptococcus*, *Actinomyces*, *Bifidobacterium*, *Corynebacterium*, *Lactobacillus*, *Eubacterium*, *Propionbacterium*, *Rothia*, *Moraxella*, *Neisseria*, *Veillonella*, *Fusobacterium*, *Eikenella*, *Heamophilus*, *Prevotella*, *Treponema* and others [40,41,42]. The data in our study demonstrated that the most prevalent bacterial genera detected in healthy dog owners from Lithuania included *Streptococcus*, *Actinomyces*, *Veillonella*, *Rothia*, *Neisseria* and *Haemophilus*. These genera are known as a part of the core oral microbiota of humans [40,42,43]. The most prevalent species in pet owners were *Actinomyces oris*, *A. massiliensis*, *Streptococcus oralis*, *S. mitis*, *S. sanguinis*, *Veilonella parvula, Pauljensenia odontolytica* and *Neisseria mucosa* with the prevalence of each species being above 3% from a total amount of microbiota. Although some of these species, for example, *S. mitis*, *S. sangunis* and *V. parvula*, were described as potential causative agents of periodontitis or other diseases [44,45], these species are a part of the normal oral microbiome in humans [46,47]. 

### 3.2. Dog Microbiome

Not many studies are associated with metagenomics on the oral microbiome of healthy dogs. According to the different sources, *Pasteurella*, *Corynebacterium*, *Capnocytophaga*, *Neisseria*, *Actinomyces*, *Porphyromonas*, *Fusobacterium* and some other genera of bacteria are predominant, forming core microbiota in dogs’ mouths and dental biofilms [28,32,48]. In our study, the largest numbers of bacteria depended on the genera *Porphyromonas, Corynebacterium, Lampropedia* and *Tannerella*, which coincides with the data obtained by other authors. In a study performed by Oba and co-authors in 2022, it was found that *Porphyromonas* and *Tannerella* were associated with poor oral health [48]. In this case, we have detected large numbers of *Porphyromonas* and *Tannerella* in dogs without signs of periodontitis or other infections. In our study, among the species of the healthy canine oral microbiome, the most prevalent species were *Porphyromonas gulae*, *P. cangigivalis*, *P. canoris*, *Corynebacterium canis*, *Tannerella forsythia* and undetermined *Lampropedia* sp. There is data that some of these species (*P. gulae*, *Tannerella forsythia*) have the potential to cause periodontitis in dogs [26,49]. Although we have selected dogs without signs of periodontitis, it is not easy to determine the status of each tooth, and there is a possibility that periodontitis could be in the development stage. As there is little knowledge about separate strains or serotypes of canine oral bacteria, it may be assumed that periodontal disease can be caused by certain types of the above-mentioned species or other factors that might determine the disease’s aetiology and manifestation. There is information that immune status, genetics, and type of feed also predispose to this pathology [50,51,52]. The data about the role of *Lampropedia* in dogs is still scarce. It was also previously detected in large numbers of supragingival plaques or other places in the mouths of healthy dogs [53,54]. 

### 3.3. Similarity of Human and Dog Microbiomes and the Influence of Keeping Dogs for Microbial Composition in Pet Owners

It is known that different lifestyles and different diets can influence microbial composition in humans. Still, there is a lack of data on how microbial communities can be influenced by animals living in close contact. Zoonoses are well-known diseases that can cause infections in a few species having contact, however, such infections affect the host rather than the microorganisms living in it. A recent study by Jiang and co-authors showed how pets influence microbial changes in elderly people. They found that dog ownership significantly modulated the composition of the gut microbiota of the dog owner [55]. In their study the abundances of *Actinobacteria*, *Bifidobacteriaceae* and *Ruminococcaceae* were significantly increased. In contrast, the abundance of *Moracellaceae* was significantly suppressed in the cohort of dogs compared with persons living without pets. The authors concluded that dog ownership can promote the increase in beneficial microorganisms and suppress the number of harmful bacteria. Our study involved young, healthy persons, and the aim was to analyse the presence of the same bacteria and AMR patterns in dogs and their owners. Even though shotgun metagenomic sequencing allowed us to obtain microbial varieties up to the strain level, we did not detect the same species in dogs and their owners. Only at a genus level were there some genera, including *Porphyromonas*, *Corynebacterium*, *Tennerella*, *Arachnia*, *Neisseria*, *Pauljensenia*, *Campylobacter*, *Actinomyces*, *Prevotella* and *Fusobacterium*, prevalent both in humans and animals. This demonstrates that it is very important to identify bacteria up to the species level, as there was no direct link between the presence of the same genera and species in dogs and their owners. In this study, substantial differences in the microbial composition of the oral microbiome were visible even at a phylum level where the most abundant bacteria in dogs and their owners depended on different phyla, with the highest prevalence of Actinobacteria and Firmicutes in humans and Bacteroidota in dogs. This demonstrates different synergistic adaptations between macroorganisms and microorganisms, either because of the evolution peculiarities of these species or the different nature of nutrition. Although the nutrition of dogs continuously changes quite drastically by reducing or excluding raw meat from dogs’ menus and feeding them processed food, we still could not see any microbial similarities between dogs and their owners. The data about dogs’ microbiome composition are diverse. Data prove that the canine oral microbiome differs significantly from the human oral microbiome, with only a 16.4% coincidence of bacterial taxa [14]. According to the study performed by Holcombe and co-authors, some bacteria, such as streptococci in dogs, are replaced by *Neisseria* spp. to compare microbial diversity with humans [56]. In our study, however, *Neisseria* in dogs had an even lower prevalence than in humans. Still, *Porphyromonas* and *Corynebacterium* have the highest prevalence, which can be compared with the level of *Streptococcus* and *Actinomyces* in their owners. In the other study performed in the Czech Republic, seventeen bacterial species occurring in pet owners and their dogs were identified. Therefore, a conclusion was made about sharing the same bacteria between dogs and their owners [57]. However, the study involved only culturable bacterial species, from which *Staphylococcus intermedius*, *Escherichia coli*, *E. faecalis*, *Acinetobacter lwoffii*, *Pseudomonas putida* and *S. aureus* were detected as common species in dogs and owners. Such different results are more likely associated with applying different methods as sequencing is based on all microbiome analysis, whereas cultivation allows isolating only a limited number of species. Nevertheless, both methods can be important depending on the aim (exploration of microbiome or detection of zoonotic culturable bacteria). 

It is known that in humans, *Porphyromonas gingivalis*, *Treponema denticola* and *Tannerella forsythia* play a key role in the aetiology of periodontal disease [58]. Interestingly, pet owners did not carry these species in our study despite their dogs being carriers of *P. gingivalis* and *T. forsythia*. Summarising the data about microbial prevalence in healthy dogs and their owners, we did not observe any of the same species in the oral microbiomes of dogs and humans. On the contrary, even though dogs carried some potential pathogens for humans, there was no detected influence of microbial migration from pets to their owners.

Our study had a limitation of relatively low numbers of tested dogs and their owners; therefore, more studies are required to prove our data that canine and human oral microbiomes are unique, and dogs living in close contact with owners had an influence on the microbial composition of humans. It is also important to mention that we have investigated only healthy individuals. Although, to date, there are no clear evidence-based studies that periodontitis pathogens of dogs can be transmitted and cause a similar disease in humans, the obtained results from a study performed by Yamasaki and colleagues suggest that several periodontopathic species could be transmitted between humans and their companion dogs [9]. Taking into account possible different bacterial compositions from healthy and diseased dogs and the possible potential risks for transmission of pathogenic species from the oral cavity of dogs, it is important to maintain good oral hygiene. This is also important from the point of animal health itself. A large-scale quantitative Swedish survey on dental care in dogs demonstrated that only 4% of Swedish dog owners brushed their dogs’ teeth daily [58]. Evidence shows that brushing dogs’ teeth is a good prophylaxis measure and prevents periodontal disease progression [58]. Knowing that dogs with periodontal disease carry different microorganisms in the oral cavity, it may be outlined that prophylaxis of oral health in dogs can reduce the risk for transmission of pathogens to other dogs, other pets and their owners. Although this was a small-scale study, it was performed together with quality control operations, such as per base sequence quality, per base N content, sequence length distribution, sequence duplication level, and overrepresented sequence controls. It included a control sample with known species for obtaining reliable results. 

### 3.4. Zoonotic Potential of the AMR Genes

We have found thirty-seven different genetic determinants in the microbiota of dogs and their owners’ encoding resistance to different classes of antimicrobials with the most prevalent resistances to β-lactams, tetracyclines and macrolides. These are the oldest used antibiotics which were used before or in the 1950s; the resistance prevalence in both humans and animals is common worldwide. Only some determinants, including *Cfx*A family broad-spectrum beta-lactamase, tetracycline, streptomycin, lincomycin and pleuromutilins ribosomal protection proteins and 23S rRNA (adenine(2058)-N(6))-methyltransferase encoding resistance to macrolides, were detected in the microbiota of both dogs and their owners. *Cfx*A beta-lactamase, in this study, was found in *Prevotella* in human microbiome and bacteria order Bacteroidales from dogs. Previous data shows that this gene was also found in *Prevotella* isolated from humans [59]. As in our study, *Prevotella* was highly prevalent in humans and it is known that this genus of bacteria is a frequent microorganism of the human oral microbiome; it may be assumed that *Cfx*A in this case originated from human microbiota. Tetracycline resistance ribosomal protection proteins, such as *tet*(O), *tet*(Q), *tet*(W) and others, demonstrated multispecies prevalence in both dogs and owners. These determinants can be found in both Gram-positive and Gram-negative bacteria of animals and humans [60]. *Tet*(32) was first detected in *Clostridium-*related human colonic anaerobes, which also carried *tet*(W) [61]. *Tet*(Q) is also previously mostly detected in the oral microbiome of humans [62,63]. In recent study, *tet*(Q) and *Cfx*A were detected in the microbiota of dogs and their owners, and the conclusion that the oral microbiota of dogs and pet owners share the same AMR genes was made [64]. In the other study based on gut metagenomics, the conclusion was made that from the perspective of families, the shared bacterial community may be the main cause of the co-occurrence of ARGs in families [65]. However, in our study, we did not find similar microbiota between the dogs and their owners. We want to note that, in our study, AMR genes’ encoding resistance to amphenicols, aminoglycosides and sulphonamides, as well as quaternary ammonium compounds, usually used in veterinary hospitals, were detected exceptionally in dogs, which demonstrates possible low resistance transfer of AMR determinants between human and canine oral microbiota. Moreover, overall, the numbers of AMR genes were higher in pet owners than in dogs. Our findings demonstrate that even though microorganisms in both dogs and their owners harbour a part of the same genes encoding AMR, the total resistome in the oral cavity of dogs and humans differs, which suggests an opinion that the risk of AMR transfer from healthy dogs to their owners is not high. Previously, we have reported the risk of AMR transfer from companion animals to humans, and the major risk factors included contamination of pets by resistant bacteria, such as methicillin-resistant *Staphylococcus aureus*, *Staphylococcus pseudintermedius*, and other culturable bacteria, prolonged hospitalisation of animals, antimicrobial usage and chronic skin infections [66]. This demonstrates different microbial compositions and AMR potential in healthy and diseased animals, as well as the importance of the site of an infection. 

## 4. Materials and Methods

### 4.1. Animals and Humans Involved in the Study

A schematic representation of the study is presented in Figure 8. 

The details about dogs and their owners involved in the study are presented in Table 2. All animals were healthy patients of the veterinary clinic located in the central Lithuania with only a history of prophylactic visiting (vaccination, general inspection, teeth inspection, neutering). All applicable international, national and institutional guidelines for the care and use of animals were followed. General health status was evaluated together with morphological and biochemical blood analysis according to the standard procedures of veterinary inspection. Oral cavities of pets were observed by veterinary odontologists. In cases of suspected periodontitis, X-rays were used to evaluate the level of periodontitis. Only healthy individuals without signs of periodontitis (PD1), and not treated for at least 6 months, were selected for further testing. The animals were selected randomly according to the above-mentioned inclusion criteria, ensuring variety in sex, age, breed, living place and neutering. Only those animals that had close contact with the owners were selected. Prior to the procedure, informed consent and signed approval were obtained from dog owners who expressed their wish for sampling together with their pets. 

The study included six dogs of both sexes, aged from 6 months to 15 years (an average of 7 years), with weights from 8 to 45 kg (an average of 28 kg), living with a single owner. Half of the dogs were mixed breeds, while the other half were pure-breed dogs. Half of the dogs were fed dry food, while the other half included a mixed food diet, which included irregular feeding with raw meat. Half of the dogs were neutered; four lived in the city or town, while the rest lived in urban areas but could freely move into the house. The dogs’ owners were adults of both sexes from 27 to 33 years who did not use antimicrobials during the last six months. All owners had close daily contact with their dogs, and four of them used to sleep in the same bed with their pets.

The sampling procedures, extracting of the DNA, sequencing and data analysis was performed simultaneously in both dogs and their owners, with the aim to obtain comparable results.

### 4.2. Sampling Procedure

Samples from dogs and their owners were obtained using sterile cotton swabs by taking dental plaque from all over the mouth for 1 min and by placing and stirring swabs in DNA/RNA Shield Collection Tubse (Zymo Research, Irvine, CA, USA) intended for sampling from the mouth without personal identification marks. Thereafter, in total, two pooled samples (one from six dogs and one from six owners) were made by mixing all the samples in equal parts into a sterile cryogenic tube and placed at −80 °C at the Microbiology and Virology Institute, Lithuanian University of Health Sciences, for further testing. Samples were then delivered to the laboratory for sequencing on ice in DNA/RNA Shield (Irvine, CA, USA).

### 4.3. Microbial Profiling and Detection of AMR

DNA isolation, quality control, library preparation and NGS were processed with the ZymoBIOMICS Shotgun Metagenomic Sequencing Service for Microbiome Analysis (Zymo Research, Irvine, CA, USA). For the isolation of the DNA the ZymoBIOMICS DNA Microprep Kit (Zymo Research, Irvine, CA, USA) was used. Sequencing libraries were prepared using Nextera^®^ DNA Flex Library Prep Kit (Illumina, San Diego, CA, USA) with up to 100 ng DNA input following the manufacturers protocol using internal dual-index 8 bp barcodes with Nextera^®^ adapters (Illumina, San Diego, CA, USA). All libraries were quantified with TapeStation^®^ (Agilent Technologies, Santa Clara, CA, USA) and then pooled in equal abundance. The final pool was quantified using qPCR. The final library was sequenced on the Illumina HiSeq (Illumina, San Diego, CA, USA). 

Raw sequence reads were trimmed to remove low quality fractions and adapters with Trimmomatic-0.33, as previously described [67]. Quality trimming by sliding window with 6 bp window size and a quality cutoff of 20, and reads with a size lower than 70 bp were removed. Antimicrobial resistance determinants were identified with the DIAMOND sequence aligner [68]. Microbial composition was profiled using sourmash [69]. The full GTDB database (R07-RS207) was used for bacterial identification. The resulting taxonomy and abundance information were further analysed: (1) to perform alpha-diversity analyses; (2) to create microbial composition barplots with QIIME [70]; (3) to create taxa abundance heatmaps with hierarchical clustering (based on Bray–Curtis dissimilarity); and (4) for biomarker discovery with LEfSe [71] with default settings (*p* > 0.05 and LDA effect size >2). Sequences were deposited at NCBI database by the access number PRJNA1010656.

### 4.4. Statistical and Data Analysis

Statistical analysis of the data on the bacterial prevalence of separate taxons between the groups was counted using a Z-Test calculator for two population proportions [72]. Results were considered statistically significant if *p* < 0.05. For the analysis of AMR genes, the genes only considered present in samples if no less than 5 copies of the same gene were detected in any of the tested groups to escape contamination or unreliable results.

## 5. Conclusions

The most prevalent bacterial genera in microbiomes of dogs without signs of periodontitis were *Porphyromonas*, *Corynebacterium*, *Lampropedia*, *Tannerella*, *Arachnia* and *Capnocytophaga*. Thirty-seven different genetic determinants were detected in the microbiota of dogs and their owners, encoding resistance to different classes of antimicrobials. The most widespread encoded resistances were detected towards β-lactams, tetracyclines and macrolides. This study demonstrated different bacterial composition in oral microbiomes of healthy dogs without clinical signs of periodontal disease and their owners. Due to the low numbers of the samples tested, further investigations with an increased number of samples should be performed to prove or to deny our findings.

## Figures and Tables

**Figure 1 antibiotics-12-01554-f001:**
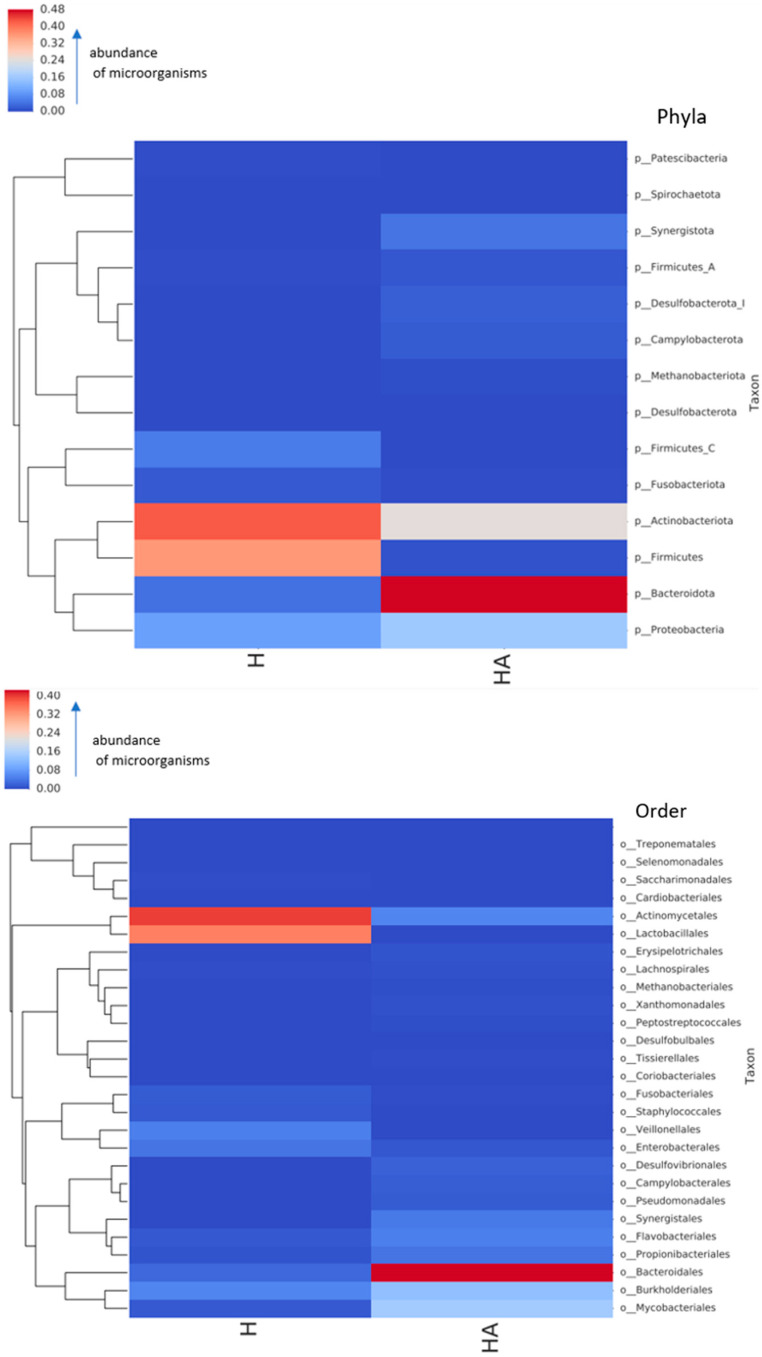
Oral microbiome composition in dogs (HA) and their owners (H) at a phylum and order level. Different colours mean different abundance of microorganisms from dark red (highest abundance) to dark blue (lowest abundance). Each row represents the abundance for each taxon, with the taxonomy ID shown on the right. Hierarchical clustering was performed on samples based on Bray–Curtis dissimilarity. Names ending with an alphabetic suffix indicate polyphyletic taxons.

**Figure 2 antibiotics-12-01554-f002:**
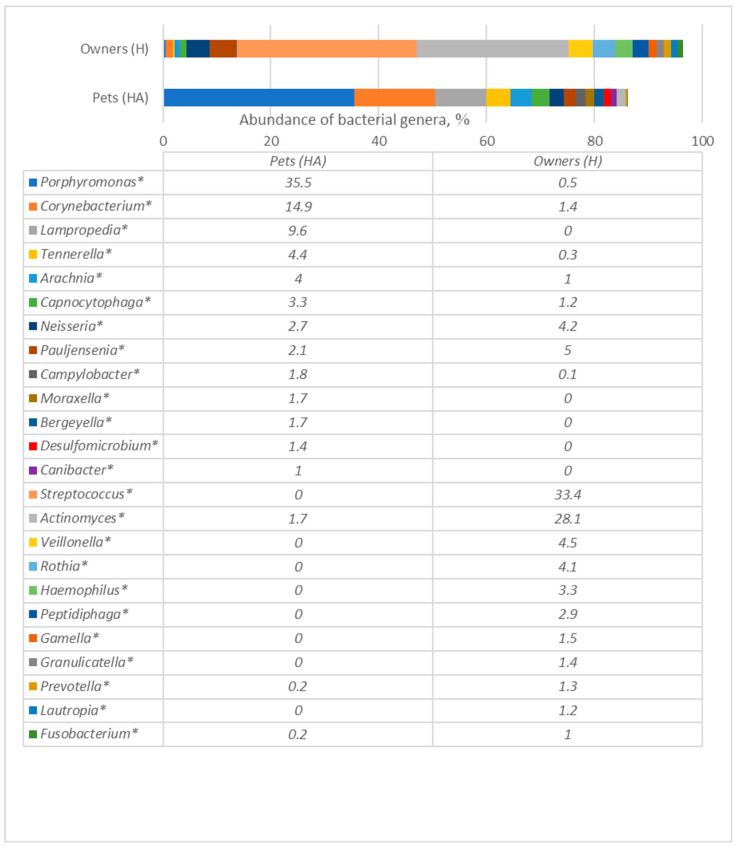
Comparison of bacterial genera of oral microbiomes of dogs (HA) and their owners (H) (only presented the genera whose prevalence was at least 1% from all bacterial amounts in any of the tested groups). The numbers in Table mean abundance (%) of separate bacterial genera. * statistically significant results between the groups.

**Figure 3 antibiotics-12-01554-f003:**
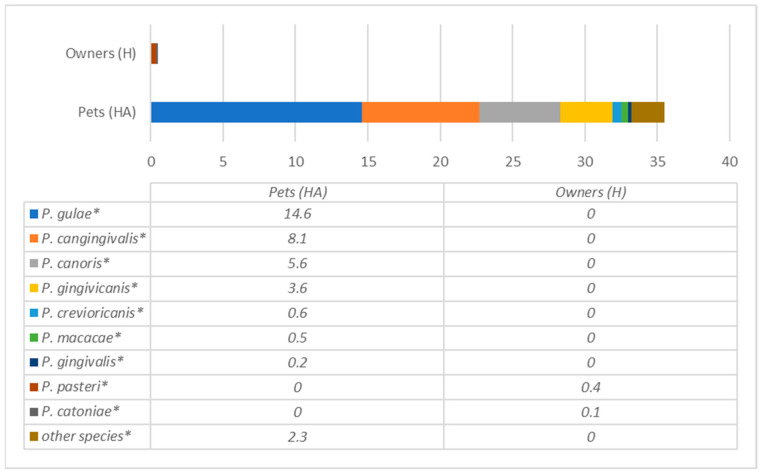
Comparison of *Porphyromonas* species prevalence (% from all bacteria) in dogs and their owners. * statistically significant results between the groups.

**Figure 4 antibiotics-12-01554-f004:**
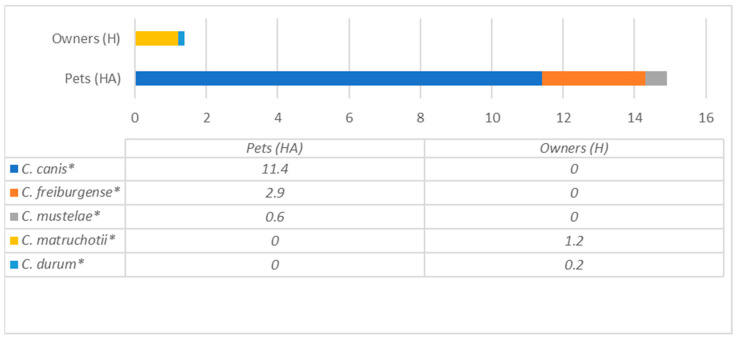
Comparison of *Corynebacterium* species prevalence (% from all bacteria) in dogs and their owners. * statistically significant results between the groups.

**Figure 5 antibiotics-12-01554-f005:**
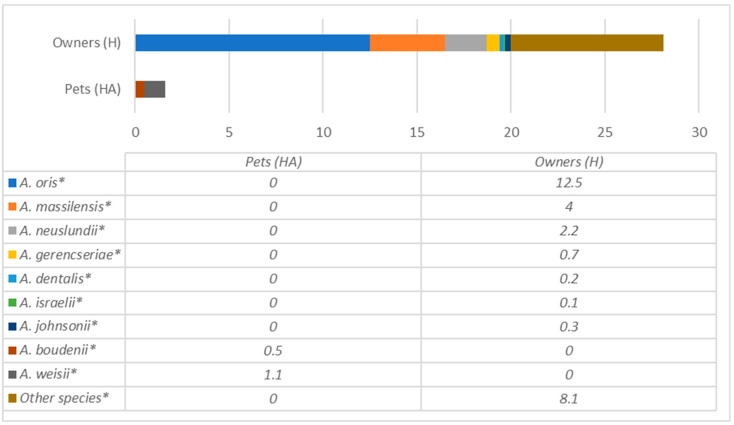
Comparison of *Actinomyces* species prevalence (% from all bacteria) in dogs and their owners. * statistically significant results between the groups.

**Figure 6 antibiotics-12-01554-f006:**
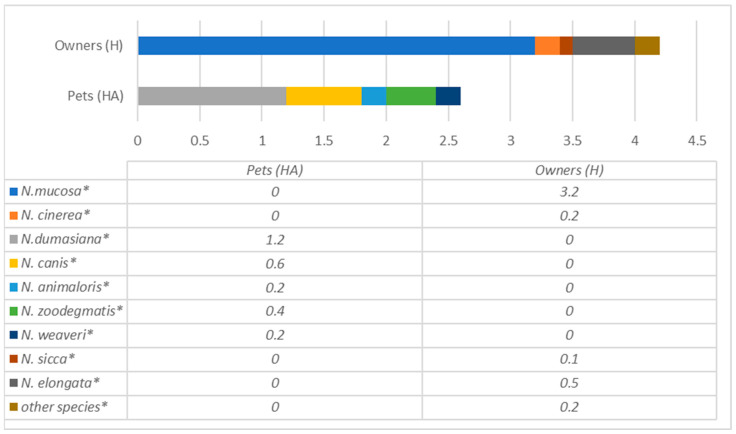
Comparison of *Neisseria* species prevalence (% from all bacteria) in dogs and their owners. * statistically significant results between the groups.

**Figure 7 antibiotics-12-01554-f007:**
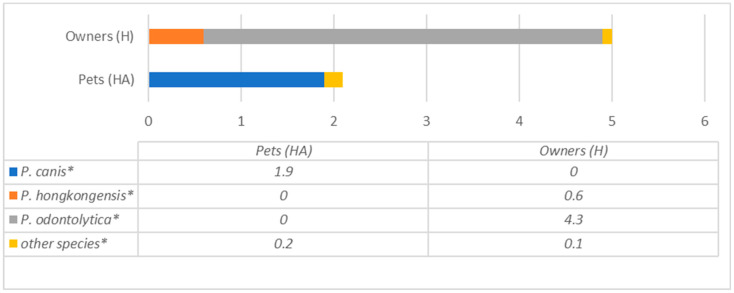
Comparison of *Pauljensenia* species prevalence (% from all bacteria) in dogs and their owners. * statistically significant results between the groups.

**Figure 8 antibiotics-12-01554-f008:**
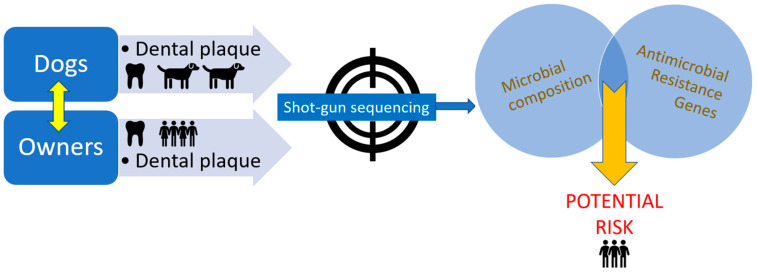
Schematic view of the study. Only healthy dogs and their owners were tested.

**Table 1 antibiotics-12-01554-t001:** AMR determinants in the microbiota of dogs and their owners.

Antimicrobial Class	Resistance Determinants *	Number of Detected Genes in Samples	
		Dogs	Owners
β-lactams	CfxA family class A broad-spectrum beta-lactamase	35	145
	class A extended-spectrum beta-lactamase CfxA4	15	0
	class A broad-spectrum beta-lactamase CfxA5	10	0
	CSP family class A extended-spectrum beta-lactamase SPU-1	0	23
	oxacillin-hydrolyzing class D beta-lactamase OXA-85	0	19
	class A extended-spectrum beta-lactamase CfxA2	0	9
	OXA-2 family class D extended-spectrum beta-lactamase OXA-539	7	0
	class A beta-lactamase BRO-1	5	0
Aminoglycosides	ANT(3″)-Ia family aminoglycoside nucleotidyltransferases	15	0
	aminoglycoside O-phosphotransferase APH(3″)-Ib	8	0
	aminoglycoside O-phosphotransferase APH(6)-Id	17	0
Tetracyclines	tetracycline efflux ABC transporter TetAB subunit A	19	0
	tetracycline efflux ABC transporter Tet(46) subunit B	4	38
	tetracycline efflux Na+/H+ antiporter family transporter Tet(35)	6	0
	tetracycline efflux ABC transporter Tet(46)	0	44
	tetracycline resistance ribosomal protection protein Tet(W)	45	305
	tetracycline resistance ribosomal protection protein Tet(M)	0	366
	tetracycline resistance ribosomal protection protein Tet(32)	131	30
	tetracycline efflux MFS transporter Tet(33)	21	0
	tetracycline resistance ribosomal protection protein Tet(Q)	36	132
	tetracycline resistance ribosomal protection protein Tet(O)	12	8
	tetracycline-inactivating monooxygenase Tet(X)	6	0
	tetracycline efflux MFS transporter Tet(Z)	0	6
	tetracycline resistance NADPH-dependent oxidoreductase Tet(37)	0	6
Amphenicols	type A-3 chloramphenicol O-acetyltransferase CatIII	9	0
	chloramphenicol efflux MFS transporter Cmx	8	0
Macrolides	ABC-F type ribosomal protection protein Msr(D)	0	1118
	23S rRNA (adenine(2058)-N(6))-methyltransferase Erm(X)	10	150
	23S rRNA (adenine(2058)-N(6))-methyltransferase Erm(39)	0	11
	23S rRNA (adenine(2058)-N(6))-methyltransferase Erm(40)	0	19
	23S rRNA (adenine(2058)-N(6))-methyltransferase Erm(F)	0	25
	macrolide efflux MFS transporter Mef(A)	0	1089
Streptogramins, pleuromutilins and lincomycin	ABC-F type ribosomal protection protein Lsa(C)	68	121
	ABC-F type ribosomal protection protein Lsa(B)	6	0
Sulfonamides	sulfonamide-resistant dihydropteroate synthase Sul1	21	0
	sulfonamide-resistant dihydropteroate synthase Sul2	12	0
Other	quaternary ammonium compound efflux SMR transporter QacE delta 1	8	0

* resistance determinants detected in bacteria of both dogs and their owners are highlighted in red.

**Table 2 antibiotics-12-01554-t002:** The main characteristics of dogs and their owners involved in the study.

Dogs							Owners			
Serial Number of Dogs	Age (Years)	Weight, kg	Breed	Diet	Living Place	Neutered	Age	Sex	Years of Keeping Dog(s)	Sleeps with Dog(s)
1.	0.5	26	Afghan Hound	dry	City	no	33	female	1	yes
2.	15	37	mixed	dry	City	no	31	male	14	no
3.	7	8	mixed	mix	urban	yes	26	female	7	yes
4.	12	28	mixed	mix	City	yes	33	female	10	yes
5.	4	25	Retriever	dry	City	yes	32	female	4	no
6.	4	45	German Shepherd	mix	urban	no	27	male	4	yes

## Data Availability

Data could be obtained from the corresponding author according to the request. Sequences were deposited at NCBI database by access number PRJNA1010656.

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
