# Peer review of "Comparison of Oral Microbial Composition and Determinants Encoding Antimicrobial Resistance in Dogs and Their Owners"

_antibiotics, 2023, doi:10.3390/antibiotics12101554_

Round 1
Reviewer 1 Report
Please find my suggestions and comments in the attached MS Word file.

The English language must be improved. Some places are really hard to follow in the abstract and in the introduction.
Author Response
Please find our response provided on a file attached.

Reviewer 2 Report
The objective of the authors was to study any potential transfer of antibiotic resistance material between dogs and humans.
This is a very interesting study, with significant novelty and public health interest, and I support publication.
Nevertheless, I have some comments, which if addressed will improve the manuscript for future reader.
Major issues
-The authors have lister the inclusion / exclusion criteria, but are missing the details of selection of these six animals, so, please add these details as well.
-Please present results of the two cohorts of dogs: those that consume only dry feed and those which only consumed raw meat.
Minor issues.
-Please clarify the details of transfer of samples to the laboratory.
-Please separate the Discussion into two sub-sections to make the flow of reading easier.
-Please add a passage regarding the clinical consequences of the findings for canine dental work.
Author Response

(The authors gave the same response as above.)

Reviewer 3 Report
Overall, the manuscript has been written well and the methods and results have been described well. I would recommend the authors to focus more on the interpretation of results and possible implications. This will make the research more relatable. Possibly, it would help to support the motivation with examples of spread of bacteria / antimicrobial genes resulting in hard-to-treat disease conditions.
Author Response

(The authors gave the same response as above.)

Reviewer 4 Report
ŠakarnytÄ— et al. in the present manuscript Comparison of oral microbial composition and determinants encoding antimicrobial resistance in dogs and their owner. They evaluated a resistant profile of microbiomes in healthy dogs and their owners. The manuscript of ŠakarnytÄ— and colleagues points towards an important and somewhat underestimated aspect of AMR. Unfortunately, the manuscript lacks the necessary controls and should be significantly revised and extended before it is suitable for publication in this journal. Specifically, the following points should be addressed:
1. The introduction should undergo significant revision and expansion to provide a more comprehensive overview.
2. The abstract needs to be more informative, conveying key points of the study.
3. The Discussion section should be expanded to offer a thorough interpretation of the findings, their significance, and potential implications.
4. Is there any diversity difference between owners and dogs in terms of characteristics or behavior?
5. Additionally, include a schematic representation of the main theme of your work.
6. Both human and animal ethical statements should be included.
7. The manuscript should be carefully proofread for proper English language usage and correctness.
need Moderate editing.
Author Response

(The authors gave the same response as above.)

Reviewer 5 Report
The authors did their research to evaluate the oral microbial composition and determinants encoding antimicrobial resistance in dogs and their owners. They conducted their research on an important topic, having zoonotic significance. However, I have a few comments as follows:
Major comments
> The major concern is the sample size. How could the authors come to a conclusion as follows with only six samples each: “it was stated that different bacterial species, low numbers of the similar AMR genes in microbiota of dogs and their owners keeping pets even in close contact, demonstrate low potential of microbial transmission between oral cavity of dogs and their owners living in a household.”
> I was just wondering how the authors could find 299 bacterial species from 6+6 samples, basically from only two. They should mention this information in both the M+M and results sections elaborately.
> Moreover, didn’t you submit your data to the NCBI or other databases? If yes, please mention their accession numbers. If you have not, please submit them, get the accession numbers, and provide the information in the manuscript.
> The manuscript has punctuation and grammatical issues. Please check and correct them throughout the manuscript.
Minor comments
Line 9: Please provide the full form of “AMR” here. Please do the same for the first use of any abbreviations.
Line 35: Please remove “According to Ghasemzedeh and Namazi” from here
Line 41-42: Eighty percent should be 80%
Table 2: You need to mention what is “f” and what is “m” as legends under the table. And please use “Serial Number of Dogs” here.
Line 341: “4” should be four. Please do the same for 1-10 throughout the manuscript.
Line 357: DNA extraction from where? Please mention it here.
The manuscript has punctuation and grammatical issues. Please check and correct them throughout the manuscript.
Author Response

(The authors gave the same response as above.)

Round 2
Reviewer 1 Report
I am generally happy with the changes the authors made, which significantly improved the manuscript. I still think that increasing the sample size of the current study will yield more interesting results, but that could be the focus of future studies.
The quality of English language was significantly improved.
Author Response
Please find file attached

Reviewer 2 Report
The authors have made extensive changes and have surely imrpoved the manuscript.
I have only last minor comment at this stage: have the authors scanned the recent literature to ensure that all relevant studies have been discussed? I have not done such a search for the last 6-8 months, so I am not up to date, but the authors can do this and include in the final manuscript any references that they might find useful for their discussion.
Then, the manuscript can be accepted.
Author Response
Please find file attached

Reviewer 5 Report
The authors addressed all my comments adequately; however, it would be better if they could increase the sample size. But I'm satisfied with the authors' clarification.
Best wishes
Author Response
Please find file attached
